# Poor psychological health and 8-year mortality: a population-based prospective cohort study stratified by gender in Scania, Sweden

Maria Fridh ,[1] Mirnabi Pirouzifard,[1,2] Maria Rosvall,[1,3] Martin Lindstrom[1,2]

[1]Social Medicine and Health Policy, Department of Clinical Sciences in Malmö, Lund University Faculty of Medicine, Malmö, Sweden
[2]Centre for Primary Health Care Research, Region Skåne and Lund University, Malmö, Sweden
[3]School of Public Health, Institute of Medicine, University of Gothenburg Sahlgrenska Academy, Goteborg, Sweden

**Correspondence to**
Dr Maria Fridh;
maria.fridh@med.lu.se

## ABSTRACT

**Objectives** We investigated gender differences in the association between mortality and general psychological distress (measured by 12-item General Health Questionnaire, GHQ-12), as an increased mortality risk has been shown in community studies, but gender differences are largely unknown.

**Setting** We used data from a cross-sectional population-based public health survey conducted in 2008 in the Swedish region of Skåne (Scania) of people 18–80 years old (response rate 54.1 %). The relationship between psychological distress and subsequent all-cause and cause-specific mortality was examined by logistic regression models for the total study population and stratified by gender, adjusting for age, socioeconomic status, lifestyle (physical activity, smoking, alcohol consumption), and chronic disease.

**Participants** Of 28 198 respondents, 25 503 were included in analysis by restrictive criteria.

**Outcome measures** Overall and cause-specific mortality by 31 December 2016.

**Results** More women (20.2 %) than men (15.7 %) reported psychological distress at baseline (GHQ ≥3). During a mean follow-up of 8.1 years, 1389 participants died: 425 (30.6%) from cardiovascular diseases, 539 (38.8%) from cancer, and 425 (30.6%) from other causes. The overall association between psychological distress and mortality risk held for all mortality end-points except cancer after multiple adjustments (eg, all-cause mortality OR 1.8 (95 % CI 1.4 to 2.2) for men and women combined. However, stratification revealed a clear gender difference as the association between GHQ-12 and mortality was consistently stronger and more robust among men than women.

**Conclusion** More women than men reported psychological distress while mortality was higher among men (ie, the morbidity-mortality gender paradox). GHQ-12 could potentially be used as one of several predictors of mortality, especially for men. In the future, screening tools for psychological distress should be validated for both men and women. Further research regarding the underlying mechanisms of the gender paradox is warranted.

## STRENGTHS AND LIMITATIONS OF THIS STUDY

⇒ This was a large, prospective, randomly selected stratified population-based study with valid exposure and outcome measures (12-item General Health Questionnaire and mortality).
⇒ The participant rate was moderately high with no substantial selection bias.
⇒ Relevant potential confounders (age, socioeconomic status, physical activity, smoking, alcohol consumption) were included in the multiple statistical models.
⇒ In order to investigate reverse causality, the final model adjusted for self-reported chronic disease, which may be a less valid indicator than objective measures of pre-existing disease at baseline.
⇒ Although statistical models included several potentially relevant confounders, residual confounding cannot be excluded.

people, is of great public health concern in Sweden.[1] Unspecified psychological distress has been shown to have a high and persistent mortality effect in the general population independently of clinical depression and other severe mental illness.[2] The mechanisms behind this are probably both biological and behavioural with intricate interactions across pathways, contributing to for example, cardiovascular disease, cancer progression and premature mortality.[3 4]

The General Health Questionnaire (GHQ) is an extensively used, reliable and validated measure of unspecific psychological distress in population studies,[5 6] and in Sweden the 12-item GHQ (GHQ-12) has been used for decades in national and regional surveys.[7] Earlier population-based research has demonstrated an association between psychological distress measured by the GHQ-12 and risk of all-cause mortality,[8–11] as well as cause-specific mortality (eg, cardiovascular death,[8 10–14] cancer[10 15 16] and other causes.[11]) The association with GHQ-12 is clear and robust for all-cause mortality and cardiovascular mortality,

## INTRODUCTION

Increasing psychological distress in the general population, especially among young

but mixed for cancer mortality (some studies show an association with higher levels of psychological distress for all-cancer mortality,[10] some for certain types of cancer only[15 16] and other studies no association at all.[11]) One confounder of the relationship between psychological distress and premature death might be the presence of pre-existing disease at baseline (contributing both to distress and subsequent higher mortality rates, ie, 'reverse causality').[10] Several studies on GHQ-12 and mortality attempted to evaluate the possibility of reverse causality by adjusting for different measures of baseline physical illness, and/or excluding all deaths the first years of follow-up. Reverse causality was not found to explain the association between GHQ-12 and all-cause mortality or cardiovascular mortality,[8 10 11 13 14] but results were less clear for cancer mortality.[10 15 16] A dose–response relationship between psychological distress and increased risk of mortality may support the existence of a causal relationship,[8] and several studies reported a dose–response pattern for all-cause mortality and cardiovascular mortality[8–11 13 14] (some very large studies showed an increased mortality risk even at lower levels of distress).[10 14] Results on dose-response for cancer mortality were less consistent.[10 16]

The fact that psychological distress is more often reported by women than men while mortality rates are higher among men is an example of the male-female health-survival paradox.[17–19] Women are in general more likely to acquire disabling conditions such as arthritis, autoimmune diseases and depression, while men are more likely to acquire lethal conditions at younger ages, such as lung cancer and myocardial infarction.[17–19] There are multiple explanations for the gender paradox, but it is thought that both sex (biological factors) and gender (social factors) play important and interacting roles.[18] The most prominent biological explanations are genetic (two X-chromosomes constitute a survival/ageing advantage), hormonal (eg, the protective effect of oestrogens on serum lipids delaying atherosclerosis in females) and immunological (testosterone may reduce the robustness of the male immunological system with a higher risk of life-threatening infections, while immunological effects of oestrogens may increase the risk of autoimmunity and chronic inflammation).[19 20] Social explanations focus on differences in behaviour and social roles. Men are more likely than women to engage in health damaging risk-taking behaviours such as excessive alcohol consumption, smoking, drug use and unsafe driving. Gender roles may also contribute to differences in willingness and ability to adopt a sick role, seek help and access healthcare (men are more likely to present late with symptoms).[18–20] Despite decades of research, it is still not fully understood whether behavioural factors explain most of the gender gap or whether biological and social differences contribute more substantially to the gender gap in health and mortality.[19]

Gender differences in the association between psychological distress measured by GHQ-12 and mortality have rarely been investigated. In our literature search, we found only one study that specifically aimed to examine gender-specific associations between GHQ-12 and all-cause mortality.[9] In this study from Finland, 923 persons (414 men and 509 women) answered GHQ-12 at baseline and were followed during a mean observation time of 11 years with 44 death events (27 men, 17 women). Adjustments were made for gender, age, socioeconomic status (SES), body mass index (BMI), smoking and physical activity. The HR increased for every GHQ-12 point. The 10-year survival for distressed (GHQ ≥4) vs non-distressed (GHQ<4) men was significantly shorter (HR 3.38 (95% CI 1.55 to 7.39)), but there was no significant difference in survival between distressed and non-distressed women. The authors concluded that the increased all-cause mortality risk associated with GHQ-12 was mainly due to excess mortality among distressed men. However, a weaker significant association among women may not have been detected due to lack of statistical power.

Using a large representative population from the Swedish region of Skåne (Scania) with baseline GHQ-12 (survey data) and 8-year prospective mortality data, this study aimed to:

1. Investigate the association between psychological distress and mortality (all-cause and cause-specific) including adjustment for self-reported chronic disease as an indicator of potential reverse causality.
2. Explore a dose–response effect of psychological distress on all-cause mortality.

Furthermore, as an additional minor aim we wanted to compare two different GHQ-12 cut-off scores: GHQ ≥3 (conventionally used in Swedish public health surveys)[7] and GHQ ≥4 (widely used internationally).[6 12 15] All analyses were performed on men and women combined and furthermore stratified by gender.

## METHODS
### Participants and study design
Data from the cross-sectional 2008 Scania public health survey were used. A total of 52 142 persons aged 18–80 years (a random stratified sample selected from the official population registers of people living in Scania including 5.8% of the total population 18–80 years old) were invited to participate. The first information letter was sent out on 27 August 2008 with the option of answering the survey online. The paper questionnaires were sent by regular mail on 5 September, as well as three reminders. A total of 28 198 persons returned completed questionnaires (response rate 54.1%). Comparison between participants with register statistics of the total population in Scania showed that the age interval 18–34 years was somewhat under-represented and the age interval 65–80 years somewhat over-represented. Some under-representation of men and persons with low formal education was also observed, but the most serious under-representation concerned people not born in Europe. To compensate for selection bias, the geographically stratified random sample was weighted by age, sex, country of birth, marital

status, income and education through a weighting variable designed by Statistics Sweden (SCB). We linked study participants to mortality data and followed them until 31 December 2016. Of the 28 198 respondents in the survey, 136 could not be traced, leaving a cohort of 28 062 persons. The study sample was restricted to those with no missing values in covariates used, that is, 25 503 individuals, 11 519 men (45%) and 13 984 women (55%).

## Patient and public involvement statement

This study was based on data from a community population survey in the Swedish region of Scania. The public was not involved in the design of this study. As the Swedish person numbers were not included in the data delivery, participants cannot be individually identified. Results of the present study will be disseminated to the public through publication in scientific journals.

## Predictor and outcome variables

Psychological distress was assessed by the GHQ-12.[5 6] The 12 items reflect different aspects of psychological health such as anxiety, depression, sleep disturbance, loss of confidence and the ability to perform daily activities and cope with everyday problems during a time-frame of 'the past few weeks'. Cronbach coefficient alpha showed good internal consistency (men: 0.893; women: 0.896). Interpretation of answers is based on a 4-point response scale scored using a bimodal method (symptom present: not at all=0, same as usual=0, more than usual=1, much more than usual=1) resulting in a GHQ-score of 0–12. Mean GHQ-score was 1.49 for men and 1.61 for women (p<0.001). We used the Swedish conventional score of ≥3 to define psychological distress.[7] To determine whether there was a dose–response relation, we categorised the results from the GHQ-12 into four groups (no distress=0; subclinical distress=1–2; moderate distress=3–5; and high distress=6–12).

Mortality was assessed from 27 August 2008 to 31 December 2016, which amounts to a maximum period of 8.34 years. (Mean value men: 8.01 years; women: 8.13 years, median value men: 8.28; women: 8.29 years). Cause of death was assessed according to ICD-10 and subdivided into three mortality categories, such as: (1) cardiovascular causes of death (eg, myocardial infarction and stroke, ICD I109–I729), (2) death from cancer (C019–C979) and (3) death from other causes (A047–B999, D329–G931, J101–869). Information regarding mortality and cause of death was obtained from The National Cause of Death Register (Dödsorsaksregistret, Socialstyrelsen).

## Covariates

The following covariates were chosen as they are both known predictors of mortality and associated with psychological distress. Age was used as a continuous variable. SES was defined by 12 categories of employment: higher non-manual; medium non-manual; lower non-manual; skilled manual; unskilled manual; self-employed/farmer; early retired (before age 65 years, for health reasons or

entitlement); unemployed; student; old-age pensioner (≥65 years of age); unclassified; and long-term sick-leave. The participants' age and SES were registry data from SCB while all other covariates were self-reported data from the Scania public health survey. Leisure-time physical activity was assessed from four alternatives, of which the lowest level, sedentary (walking, bicycling, etc less than 2 hours per week) was dichotomised as 'yes' and all other options as 'no'. Smoking was assessed by three categories: daily; intermittent/non-daily; non-smoker, of which daily smoking was dichotomised as 'yes' and all other options as 'no'. Frequency of alcohol consumption past year was assessed by five categories. BMI was based on self-reported data and did not differ between distressed and non-distressed men and women, respectively. As there was no significant age-gender adjusted association between BMI and mortality in our study sample, BMI was excluded from the multiadjusted analyses. Chronic disease was assessed by the question: 'Do you have any long-term disease, injury-related trouble, disability or other weakness?' (yes/no).

## Statistics

$\chi^2$ and t-tests were used to compare the sample characteristic differences between non-distressed/distressed, men and women respectively (cut-off GHQ 2/3) (table 1).

As the main aim of the study was to test gender differences, an initial test for effect modification by gender was performed. The interaction term between psychological distress and sex was significant (OR=0.5; p=0.008) indicating that the effect of psychological distress on mortality was different for men and women.

The p value for proportionality based on the interaction term between the variable psychological distress and time of follow-up was <0.001 indicating non-fulfilment of the proportional hazards assumption. Logistic regression models were used to compute study specific ORs with accompanying 95% CI for the association between distress and all-cause mortality, as well as the association between distress and death caused by cardiovascular disease, cancer or other causes, respectively. Model 0 adjusted for age and gender, model 1 furthermore adjusted for SES, low physical activity, daily smoking and alcohol consumption, and model 2 furthermore adjusted for chronic disease. Table 2 shows logistic regression models for women and men combined. Table 3 shows logistic regression models stratified by gender (model 0 age-adjusted only). Table 4 shows overall and gender-stratified associations between different levels of psychological distress and all-cause mortality (GHQ score: 0, 1–2, 3–5 and 6–12). The statistical analyses were performed using SAS V.9.4, and all analyses were performed on weighted data. To ensure appropriate variance estimation on weighted data, we used bootstrap methods with 2000 numbers of replicates to obtain CIs and p values. The survey means procedure was used to obtain weighted descriptive statistics for continuous variables, and the survey frequency procedure was used for weighted one-way and multiway crosstabulations.

**Table 1** Descriptive characteristics (%) of age, SES, sedentary leisure time, daily smoking, alcohol, BMI and chronic distress by psychological distress (GHQ-12 ≥3)

| | Women n=13 984 | | | Men n=11 519 | | |
|---|---|---|---|---|---|---|
| | Psychological distress | | | Psychological distress | | |
| | No | Yes | | No | Yes | |
| | n=11 452 | n=2532 | | n=9925 | n=1594 | |
| | 79.8% | 20.2% | p value | 84.3% | 15.7% | p value |
| Age, years: mean±SD* | 46.9±15.9 (46.5 to 47.3) | 40.6±16.6 (39.8 to 41.4) | <0.001 | 47.5±17.6 (47.0 to 47.9) | 43.4±17.5 (42.4 to 44.3) | <0.001 |
| BMI mean±SD* | 24.9±4.6 (24.8 to 25.1) | 24.9±5.2 (24.7 to 25.2) | 0.947 | 26.1±4.1 (26.0 to 26.2) | 26.3±5.0 (26.0 to 26.6) | 0.165 |
| Socioeconomic status (SES)† | | | | | | |
| Higher non-manual | 7.8 (7.3 to 8.4) | 7.2 (6.0 to 8.3) | | 9.9 (9.2 to 10.6) | 9.0 (7.3 to 10.7) | |
| Medium non-manual | 15.9 (15.1 to 16.7) | 13.6 (11.9 to 15.2) | | 12.3 (11.5 to 13.2) | 9.3 (7.8 to 10.9) | |
| Lower non-manual | 10.6 (10.0 to 11.3) | 9.6 (8.2 to 11.0) | | 5.3 (4.7 to 5.9) | 5.3 (3.9 to 6.6) | |
| Skilled manual | 9.6 (8.9 to 10.3) | 7.7 (6.4 to 9.0) | | 11.8 (11.0 to 12.6) | 11.3 (9.3 to 13.3) | |
| Unskilled manual | 12.4 (11.6 to 13.2) | 11.5 (10.0 to 12.9) | | 13.5 (12.6 to 14.4) | 9.8 (8.1 to 11.2) | |
| Self-employed/farmer | 4.1 (3.7 to 4.6) | 3.2 (2.3 to 4.0) | | 8.4 (7.8 to 9.1) | 6.1 (4.8 to 7.4) | |
| Early retired | 3.9 (3.5 to 4.3) | 6.7 (5.5 to 7.9) | | 2.6 (2.2 to 3.0) | 8.7 (6.9 to 10.6) | |
| Unemployed | 3.1 (2.6 to 3.6) | 8.8 (7.3 to 10.2) | | 2.9 (2.4 to 3.3) | 9.3 (7.4 to 11.2) | |
| Student | 8.5 (7.8 to 9.2) | 13.5 (11.7 to 15.4) | | 6.4 (5.7 to 7.1) | 9.8 (7.8 to 11.7) | |
| Old age pensioner | 19.1 (18.2 to 19.9) | 9.7 (8.3 to 11.0) | | 20.3 (19.5 to 21.2) | 10.0 (8.4 to 11.5) | |
| Unclassified | 4.2 (3.6 to 4.7) | 4.6 (3.5 to 5.7) | | 6.1 (5.4 to 6.8) | 7.9 (6.0 to 9.7) | |
| Long-term sickleave | 0.7 (0.5 to 0.9) | 4.0 (3.0 to 5.0) | <0.001 | 0.4 (0.3 to 0.6) | 3.5 (2.5 to 4.6) | <0.001 |
| Sedentary leisure time† | 11.0 (10.3 to 11.8) | 19.8 (17.8 to 21.8) | <0.001 | 13.4 (12.6 to 14.3) | 27.1 (24.3 to 29.9) | <0.001 |
| Daily smoking† | 14.6 (13.8 to 15.4) | 19.2 (17.2 to 21.2) | <0.001 | 12.5 (11.6 to 13.3) | 21.0 (18.3 to 23.7) | <0.001 |
| Alcohol drinking past year† | | | | | | |
| Never | 14.0 (13.1 to 14.8) | 18.6 (16.7 to 20.6) | | 8.6 (7.9 to 9.3) | 13.6 (11.3 to 15.9) | |
| Once a month or more seldom | 27.2 (26.2 to 28.2) | 27.8 (25.6 to 30.0) | | 18.2 (17.2 to 19.1) | 22.7 (20.0 to 25.3) | |
| 2–4 times a month | 34.4 (33.3 to 35.5) | 32.8 (30.6 to 35.1) | | 37.7 (36.5 to 38.8) | 32.3 (29.4 to 35.2) | |
| 2–3 times a week | 20.0 (19.1 to 20.9) | 15.2 (13.5 to 16.9) | | 26.1 (25.0 to 27.1) | 22.0 (19.4 to 24.5) | |
| At least four times a week | 4.5 (4.0 to 4.9) | 5.6 (4.6 to 6.6) | <0.001 | 9.5 (8.9 to 10.2) | 9.4 (7.8 to 11.1) | <0.001 |

Continued

**Table 1** Continued

| | Women | | | Men | | |
|---|---|---|---|---|---|---|
| | n=13 984 | | | n=11 519 | | |
| | Psychological distress | | | Psychological distress | | |
| | No | Yes | | No | Yes | |
| | n=11 452 | n=2532 | | n=9925 | n=1594 | |
| | 79.8% | 20.2% | p value | 84.3% | 15.7% | p value |
| Chronic disease†‡ | 26.5 (25.5 to 27.4) | 43.1 (40.7 to 45.5) | <0.001 | 24.3 (23.2 to 25.3) | 45.5 (42.4 to 48.5) | <0.001 |

The 2008 public health survey of Scania, Sweden. Total population n=25 503. Weighted prevalence.
GHQ-12=Twelve-item version of the GHQ, 0–12 points. Psychological distress was defined as GHQ-12 ≥3.
The values in parentheses are 95% CI for mean or per cent based on bootstrap method with 2000 number of replicates.
*P value: Independent samples t-test, two-tailed.
†P value: Pearson $\chi^2$ test, two sided.
‡Chronic disease=long-term disease, injury-related trouble, disability or other weakness.
BMI, body mass index; GHQ-12, 12-item General Health Questionnaire; SES, socioeconomic status.

The Surveylogistic was used for weighted logistic regression analysis.[21] In order to compare two different cut-off scores, identical statistical analyses using cut-off GHQ 3/4 were furthermore performed (see online supplemental tables 1–4).

## RESULTS

At baseline, more women than men were psychologically distressed, 20.2% compared with 15.7% (table 1). Distressed study members were in average younger, more likely to not belong to the workforce, more likely to be sedentary, to smoke and to have a chronic disease. As can be seen by non-overlapping CI, distressed women were significantly younger, leaner, less sedentary and drank alcohol less often than distressed men.

The mortality rates (# of deaths per 100 person-years) were calculated for men and women (data not shown).

The overall mortality rate was 0.88 (95% CI 0.82 to 0.95) for men and 0.50 (95% CI 0.47 to 0.55) for women, with a male to female incidence rate ratio (IRR) of 1.75 (95% CI 1.57 to 1.95; p<0.001). The male to female IRR for cause-specific mortality were as follows: cardiovascular causes 2.51 (95% CI 2.05 to 3.07; p<0.001), cancer 1.34 (95% CI 1.13 to 1.59; p<0.001) and 'other causes' 1.74 (95% CI 1.44 to 2.11; p<0.001), with non-overlapping 95% CIs for men and women, respectively.

Table 2 shows associations between psychological distress and mortality (all-cause and cause-specific) in multiadjusted logistic regression analyses for men and women combined. Age-gender adjusted associations remained significant through model 1 (SES, physical activity, smoking, alcohol) and model 2 (chronic disease) for all mortality end-points except cancer. For all-cause mortality, the initial age-gender

**Table 2** ORs from logistic regression models for all-cause mortality and cause-specific mortality, showing association with psychological distress (GHQ≥3)

| Cause of death | Model 0 | | Model 1 | | Model 2 | | No of deaths |
|---|---|---|---|---|---|---|---|
| | OR | 95% CI | OR | 95% CI | OR | 95% CI | |
| All causes | 2.8*** | 2.3 to 3.4 | 2.1*** | 1.7 to 2.6 | 1.8*** | 1.4 to 2.2 | 1389 |
| Cause-specific: | | | | | | | |
| Cardiovascular | 2.9*** | 2.1 to 4.1 | 2.2*** | 1.5 to 3.1 | 1.8*** | 1.3 to 2.6 | 425 |
| Cancer | 1.7*** | 1.3 to 2.3 | 1.5* | 1.1 to 2.0 | 1.3 | 1.0 to 1.8 | 539 |
| Other causes | 2.8*** | 2.0 to 3.8 | 1.9*** | 1.4 to 2.6 | 1.6** | 1.2 to 2.2 | 425 |

Men and women combined; n=25 503.
The 2008 Scania public health survey, with 8.3 years follow-up.
Model 0 adjusted for age and gender.
Model 1 furthermore adjusted for socioeconomic status, physical activity, smoking and alcohol.
Model 2 furthermore adjusted for chronic disease.
Weighted OR. Bootstrap method (2000 replicates) for variation estimation.
Bold values represent statistically significant OR.
*p<0.05, **p<0.01, ***p<0.001.
GHQ, General Health Questionnaire.

**Table 3**  ORs from logistic regression models for all-cause mortality and cause-specific mortality, showing gender-stratified associations with psychological distress (GHQ≥3)

| Cause of death | Model 0 | | Model 1 | | Model 2 | | No of deaths |
|---|---|---|---|---|---|---|---|
| | OR | 95% CI | OR | 95% CI | OR | 95% CI | |
| All causes | | | | | | | |
| Women | **2.3***** | 1.7 to 3.0 | **1.6**** | 1.2 to 2.1 | **1.4*** | 1.0 to 1.8 | 574 |
| Men | **3.3***** | 2.5 to 4.4 | **2.6***** | 1.9 to 3.5 | **2.1***** | 1.6 to 2.9 | 815 |
| Cause-specific: | | | | | | | |
| Cardiovascular | | | | | | | |
| Women | **2.5***** | 1.5 to 4.2 | 1.7 | 1.0 to 2.8 | 1.4 | 0.8 to 2.4 | 140 |
| Men | **3.1***** | 2.1 to 4.8 | **2.5***** | 1.5 to 4.0 | **2.1**** | 1.3 to 3.3 | 285 |
| Cancer | | | | | | | |
| Women | **1.5*** | 1.0 to 2.3 | 1.2 | 0.8 to 1.9 | 1.1 | 0.7 to 1.7 | 258 |
| Men | **2.0***** | 1.3 to 3.0 | **1.7*** | 1.1 to 2.7 | 1.5 | 1.0 to 2.3 | 281 |
| Other causes | | | | | | | |
| Women | **2.4***** | 1.6 to 3.8 | **1.6*** | 1.0 to 2.5 | 1.3 | 0.8 to 2.1 | 176 |
| Men | **3.0***** | 2.0 to 4.5 | **2.2***** | 1.4 to 3.3 | **1.8*** | 1.1 to 2.8 | 249 |

Stratified by gender; n=13 984 women and 11 519 men.
The 2008 Scania public health survey, with 8.3 years follow-up.
Model 0 adjusted for age.
Model 1 furthermore adjusted for socioeconomic status, physical activity, smoking and alcohol.
Model 2 furthermore adjusted for chronic disease.
Weighted OR. Bootstrap method (2000 replicates) for variation estimation.
Bold values represent statistically significant OR.
*p<0.05, **p<0.01, ***p<0.001.
GHQ, General Health Questionnaire.

adjusted OR 2.8 (95 % CI 2.3 to 3.4) decreased to 1.8 (95 % CI 1.4 to 2.2) in the final model.

Gender stratification revealed a clear difference (table 3). Among men the association between psychological distress and mortality remained robust through the modelling for all mortality-outcomes except cancer. Among women, the association was only significant through the modelling for all-cause mortality. The age-adjusted OR for all-cause mortality decreased from 3.3 (95 % CI 2.5 to 4.4) to 2.1 (95 % CI 1.6 to 2.9) in men, and from 2.3 (95 % CI 1.7 to 3.0) to 1.4 (95 % CI 1.0 to 1.8) in women after full adjustments.

In order to investigate a dose–response pattern in the association between GHQ-12 and all-cause mortality, the study population was divided into four groups according to GHQ-score: 0 no distress=reference category (69%), 1–2 (15%), 3–5 (8%) and 6–12 (8%). A clear pattern of increasing mortality by increasing distress was seen for men and women combined in the fully adjusted logistic regression analyses (table 4). Gender stratification showed that this was mainly due to a strong dose-response pattern among men.

**DISCUSSION**

This study showed that the association between psychological distress measured by GHQ-12 and 8-year mortality was stronger and more robust in men than women. Among men, the association held through all adjustments (SES, lifestyle factors and chronic disease) for all-cause mortality, cardiovascular mortality and mortality from other causes except cancer. Among women, the association held through all adjustments for all-cause mortality only. These results are concordant with those of an earlier Finnish longitudinal study on GHQ-12 and 11-year all-cause mortality, that is, the increased mortality risk was mainly explained by excess mortality among distressed men.[9] As previously shown, the association between GHQ-12 and mortality was weakest for cancer.[10 11] Evidence of reverse causality was not found except possibly for cancer, which is in line with previous research.[10]

The gender morbidity-survival paradox was illustrated in this study, as morbidity (psychological distress) was higher among women, while mortality was higher among men. Several potential explanations could be considered. To begin with, the definition of psychological distress could be gendered per se,[22] as expressions of mental ill health may differ between men and women.[17] Men's and women's overall mental health could be similar, although expressed as different conditions—depression may be thought of as a female problem because more women seek help for depression, but more men commit suicide and turn to drinking and drug-abuse to alleviate symptoms of mental ill health.[17] The GHQ-12 asks for symptoms of anxiety, depression, sleep disturbance, loss of confidence and the ability to perform daily activities and cope with

**Table 4** Associations between different levels of psychological distress (GHQ-12) and all-cause mortality

| GHQ-score | Model 0 OR | 95% CI | Model 1 OR | 95% CI | Model 2 OR | 95% CI | No of deaths |
|---|---|---|---|---|---|---|---|
| **Total population** | | | | | | | 1389 |
| 0=REF | 1.0 | | 1.0 | | 1.0 | | |
| 1–2 | **1.8***** | 1.4 to 2.2 | **1.5***** | 1.2 to 1.9 | **1.4**** | 1.1 to 1.7 | |
| 3–5 | **2.4***** | 1.8 to 3.2 | **1.9***** | 1.4 to 2.6 | **1.6**** | 1.2 to 2.1 | |
| 6–12 | **3.8***** | 3.0 to 5.0 | **2.7***** | 2.0 to 3.5 | **2.2***** | 1.6 to 2.8 | |
| **Women** | | | | | | | 574 |
| 0=REF | 1.0 | | 1.0 | | 1.0 | | |
| 1–2 | **1.8***** | 1.3 to 2.4 | **1.4*** | 1.0 to 2.0 | 1.3 | 0.9 to 1.8 | |
| 3–5 | **2.1***** | 1.5 to 3.2 | **1.6*** | 1.1 to 2.4 | 1.4 | 0.9 to 2.0 | |
| 6–12 | **2.9***** | 2.0 to 4.1 | **1.9***** | 1.3 to 2.7 | **1.6*** | 1.1 to 2.2 | |
| **Men** | | | | | | | 815 |
| 0=REF | 1.0 | | 1.0 | | 1.0 | | |
| 1–2 | **1.8***** | 1.3 to 2.4 | **1.5**** | 1.1 to 2.1 | **1.4*** | 1.0 to 1.9 | |
| 3–5 | **2.6***** | 1.7 to 3.8 | **2.1***** | 1.4 to 3.3 | **1.8**** | 1.2 to 2.7 | |
| 6–12 | **4.7***** | 3.3 to 6.7 | **3.4***** | 2.3 to 5.0 | **2.7***** | 1.8 to 4.0 | |

Total population (n=25 503) and stratified by gender (n=13 984 women and 11 519 men).
The 2008 Scania public health survey, with 8.3 years follow-up.
Model 0 adjusted for age (and gender in analysis of total population).
Model 1 furthermore adjusted for socioeconomic status, physical activity, smoking and alcohol.
Model 2 furthermore adjusted for chronic disease.
Weighted OR. Bootstrap method (2000 replicates) for variation estimation.
Bold values represent statistically significant OR.
*p<0.05, **p<0.01, ***p<0.001.
GHQ, General Health Questionnaire.

everyday problems. Men may find it more difficult than women to recognise and report symptoms involving vulnerability because of masculine gender norms,[22] with an ensuing higher threshold to report psychological symptoms on questionnaires. Consequently, the men who do report symptoms might have a more severe disorder as suggested by their heightened mortality risk.[23] Gender is a ubiquitous aspect of mental health, present in contexts ranging from formal diagnostic criteria to the ways individuals label, communicate and cope with their problems.[22] A more ruminative coping style among women may contribute to an increased vulnerability to depression, but women are also more willing to adopt a sick role, seek help and access healthcare, which facilitates earlier detection.[20 24] Clinical depression is about twice as common in women as in men.[24] Sex differences in depression begin during early adolescence and are most pronounced during the reproductive years, and a possible biological explanation to female susceptibility to depression could be associated with oestrogen and progesterone.[17 24] The heritability of depression is also greater in women than men. Sex differences in monoamine functioning have been demonstrated (eg, a decrease in serotonin transmission increased depression significantly more in women than men), and women also respond better to antidepressive pharmacological treatment with selective serotonin reuptake inhibitors (SSRI) than men.[24] Furthermore, economic explanations and environmental factors may contribute to the gender difference in depression prevalence, such as a female disadvantage of lower SES, gender pay gap, strain from dual (work–family) roles and higher rates of exposure to IPV (intimate partner violence).[18 24]

A recent study investigated psychological distress and heart disease mortality in the USA, with data from more than half a million participants in the National Health Interview Survey and 18 years of mortality follow-up.[25] Results showed a robust dose–response relationship between psychological distress (measured by Kessler 6-Item Psychological Distress Scale, K-6) and heart disease mortality, indicating psychological distress to be an important predictor of heart disease mortality. The relative risk of heart disease mortality associated with psychological distress was higher among men than women, in line with results from the present study.[25] Another study highlighted the importance of identifying men with psychological distress when assessing CVD risk, as psychological distress was associated with higher cardiovascular risk scores among men but not women.[26] Psychological distress may increase the risk of heart disease through various mechanisms, for example, promoting inflammatory processes and activating automatic nerve system or hypothalamic–pituitary–adrenal influences.[25]

Psychological distress is also highly correlated with traditional CVD risk factors such as smoking, physical inactivity, obesity, hypertension and non-adherence to medical treatment.[12 25] Further research is needed to address pathways through which psychological distress increases mortality and how different underpinning pathways will be responsible for gendered patterns of morbidity and mortality in different health outcomes.[18 19 25] The underlying mechanisms in gender differences need to be better understood in order to develop effective public-health and clinical interventions. Our study supports the validity of GHQ-12 as a predictor of mortality, primarily among men. It has been suggested that GHQ-12 might possibly be an even stronger predictor of mortality than CIDI-diagnosed mood/anxiety disorders, since diagnosed mental disorders tend to be treated whereas self-reported psychological distress does not.[11] In the future, it would be valuable to validate screening tools for psychological distress separately for men and women.[24]

A minor additional aim of the present study was to compare two different GHQ-12 cut-off scores: GHQ-12 ≥3 (conventionally used in Sweden) and GHQ-12 ≥4 (widely used internationally). The choice of a cut-off point depends on the purpose of the study, and a recent Swedish case–control study concluded that although the cut-off points 2–4 were all deemed acceptable, the best sensitivity and specificity of GHQ-12 when discriminating between healthy controls and psychiatric outpatients was seen at cut-off ≥4.[7] In this study, GHQ-12≥3 defined 17.9% of the study population as psychologically distressed (women: 20.2%; men: 15.7 %), while GHQ-12 ≥4 defined 14.2% as psychologically distressed (women: 16.2%; men: 12.1 %) (online supplemental table 1). The higher cut-off led to generally slightly stronger associations with mortality (online supplemental tables 2 and 3). The higher cut-off also led to a slightly more pronounced dose-response pattern for all-cause mortality in men but a less consistent pattern in women (online supplemental table 4).

### Strengths and limitations

This was a large prospective community study based on a stratified randomised sample of people in Scania 18–80 years old who participated in the 2008 Scania Public Health Survey. The response rate of 54% is a limitation, but a weighting variable was specifically designed to compensate for self selection bias in the statistical analyses. Non-responders were more often young, male, low-educated or foreign-born. We do not know if non-responders had more psychological distress than responders, but there were probably more non-responders among persons with serious psychiatric disease (such as psychosis, schizophrenia, severe depression who are known to have a higher risk of premature death[27]), due to the mental task involved in answering a large survey with 134 main questions (totalling 273 items including subqueries and follow-up questions). Most base line data were self-reported and our study lacked objective measures of health such as blood pressure,

verified physical/mental illness and biomarkers. Among those who died in our study population, more than 50% reported baseline chronic disease (data not shown). We had no objective measure of pre-existing disease at baseline, and self-reported chronic disease may not be a valid indicator of pre-existing disease. The mortality rates across age strata were similar to national data.[28] Psychological distress was measured at baseline only, and the persistence of distress may be an important determinant of mortality risk.[29] With an instrument such as the GHQ-12 asking for symptoms during 'the past few weeks' a smaller portion of persons with short-term/intermittent than long-term/persistent psychological distress will be identified at baseline. Over time, some of the participants defined as cases at baseline will recover while some defined as non-cases will develop psychological distress, a form of misclassification diluting the effect of psychological distress on mortality over time (length-time bias).[13 30 31] This phenomenon was clearly illustrated by a large prospective study investigating the temporal robustness of psychological distress (Kessler 6 items, K-6) by re-estimating the mortality impact at 2-year, 5-year and 10-year follow-up times.[25] Data used in the present study did not fulfil the proportional assumption. Logistic regression analysis was therefore used instead of cox regression survival analysis, the method of choice in previous comparable studies.[2 8–16 23 25 29] However, additional analyses showed that the results from Cox regression analysis and logistic regression analysis were fairly similar in magnitude (data not shown).

### Conclusions

This study extends previous research by investigating gender differences in the association between psychological distress measured by GHQ-12 and mortality. Results showed strong and robust associations in men but not in women. GHQ-12 could be used as one of several predictors of mortality, especially among men. The results of the present study are probably mainly generalisable to other western countries, as both exposure and outcome are highly internationally validated measures, but the mediating factors between psychological distress and mortality may be more culturally diverse.

**Acknowledgements** The authors wish to thank all participants for their cooperation, and Region Skåne (Scania) for providing data from the 2008 Scania public health survey.

**Contributors** ML and MF designed the study in collaboration with MR. ML assembled the database. MP analysed the data with input from ML and MF. MF wrote the first draft. All authors contributed to the paper and approved the final draft. MF is the author acting as guarantor.

**Funding** This work was supported by the Swedish Research Council (Vetenskapsrådet) (K2014-69X-22427-01-4) and the Swedish ALF Government (Grant Dnr F 2014/354).

**Competing interests** None declared.

**Patient and public involvement** Patients and/or the public were not involved in the design, or conduct, or reporting, or dissemination plans of this research.

**Patient consent for publication** Not applicable.

**Ethics approval** This study involves human participants and was approved by The Regional Ethical Committee in Lund, SwedenEthics name ID: Dnr 2010/343. Participants gave informed consent to participate in the study before taking part.

**Provenance and peer review** Not commissioned; externally peer reviewed.

**Data availability statement** All data relevant to the study are included in the article or uploaded as online supplemental information. Not applicable.

**ORCID iD**
Maria Fridh http://orcid.org/0000-0002-7133-9971

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
