## [Reviewer comments · BMJ Open]

ARTICLE DETAILS

TITLE (PROVISIONAL)	Poor psychological health and 8-year mortality: A population-based prospective cohort study stratified by gender in Scania, Sweden
AUTHORS	Fridh, Maria; Pirouzifard, Mirnabi; Rosvall, Maria; Lindstrom, Martin

VERSION 1 – REVIEW

REVIEWER	Lundin, Andreas Karolinska Institute, Department of public health sciences
REVIEW RETURNED	28-Jan-2019

GENERAL COMMENTS	This is a well conducted study which examines the association between psychological distress/General health questionnaire scores and mortality. The survey used is large and population-based and linked to national registers of cause of death. The methods used are standard in epidemiology and the survival analysis is presented in a transparent way. I have some suggestions which the authors may wish to address: 1. The GHQ-12 predicts all-cause mortality, and also death from cancer and CVD as expected from previous literature. This could be taken as evidence of predictive validity of the GHQ-12 in this setting, and because of the gender stratified analyses, that this is invariant across sex. What I think is missing is a sentence or two explaining why a measure of psychological distress would lead to these specific outcomes, rather than say depression or suicide. A mechanism could be presented for this (in the introduction).2. The authors make the sex stratified analysis an important feature of their article, but this is not reflected in the title.3. Methods: A design weight corrected for non-participation is presented, but a bit too brief. I suggest the construction of the weights is presented in a bit more detail. The strata are geographical areas (municipalities and city areas) so a weight to compensate for different sample probabilities for these are probably the first weight. This is the corrected for non-participation by using register information on all those in the sample frame. Relatedly, the use of this weight should be mentioned in the statistical analysis section.4. Last sentence in participants section: those 136 most likely emigrated or were dead? Please clarify.5. Under description of the GHQ-12 you describe that scores were used to construct zones. Could you describe why these specific cut offs were used? Here it would also be good to add Cronbach coefficient alpha for the GHQ-12, separately for men and women to show if the scale had the same properties for men and women. Any differences in predictive ability could be due to gender invariance.6. Discussion: comparison with previous literature is made with Finland. Finland is not a Scandinavian county as stated.7. Tables: tables in the text are good. The tables following the article, numbered in a similar way seem to test a different cut off score (four or more symptoms). Is this correct?
--

REVIEWER	Ponizovsky, Alexander M. Israeli Ministry of Health
REVIEW RETURNED	03-Feb-2019

GENERAL COMMENTS	This study using the representative sample of the general population in the Swedish region of Scania have replicated previous findings on the well-established relationships between the GHQ-12 psychological distress scores and all-cause and specific-cause mortality rates. In addition, gender differences were shown in the psychological distress-mortality relationships with higher distress scores in women, but higher mortality rates in men. The study's methodology is adequate, although the moderating/mediating effects of gender on the psychological distress-mortality association would be additionally explored. Results of the study are presented clearly. Limitations of the study are generally noted. Because the main aim of the study was declared as to indicate gender differences in the distress-mortality relationships, the Introduction section needs more theoretical background on biological/psychological/social differences in psychological morbidity and distress expression in men and women. Also, in the Discussion section, it is absolutely insufficient to say the phrase "Among the possible explanations of the gender paradox are differentials between the sexes in biological risk, in health behaviors and social roles, and in health seeking behavior [16, 22]. "The possible explanations" should be not only listened, but also discussed in details and they should be supported by the findings of previous studies and the above-mentioned theoretical background.
--

REVIEWER	Chiu, Maria Institute for Clinical Evaluative Sciences, Research
REVIEW RETURNED	07-Feb-2019

GENERAL COMMENTS	Thank you for the opportunity to review this interesting and well-written paper entitled, "Psychological distress measured by GHQ-12 and mortality: A 5-year prospective population-based study in Scania, Sweden." The aim of this paper was to investigate whether the association between psychological distress and mortality differed between men and women in Scania, Sweden. The authors studied 28198 adults followed for up to 5 years and found that those who reported psychological distress at baseline had a higher hazards of dying than those with no psychological distress and this effect differed between men and women. They describe a morbidity-mortality gender paradox, in which women had higher prevalence of psychological distress but men with psychological distress have a higher risk of mortality. Major comments (*most important to address): *1A. Methods: It is not clear whether weighted analyses were performed for this study. If the intention was to generalize these findings to the population of Scania and adjust at least in part for differential survey sampling, then weighted analyses would be recommended. Specifically, all statistical tests (i.e. t-tests, chi-square tests, interaction testing, Cox PH, etc.) would have to be weighted. *1B. Methods: Furthermore, the authors should describe what methods were used to ensure that appropriate variance estimations
--

were performed with weighted data (e.g. bootstrapped p-values, 95% confidence intervals?)

2. Methods: What did you mean by "The differences between unweighted and weighted data were small". Which variables were examined?

*3. Statistics: If the main aim of the study was to test gender differences, was there an initial statistical test for effect modification by gender? I don't doubt that there's an interaction, but would be good to describe in methods and state that the interaction term was significant in model 0, 1, 2 (p=)?

4. In the Results, it would be good to add more details describing findings from Table 1; i.e. factors that are present in those with psychological distress in men vs. women.

*5. One major concern in the interpretation of the gender differences is whether they are statistically different. The 95% CIs for HRs, for example, overlap between men and women. I strongly suggest calculating and reporting the *mortality rates* (# of deaths per 100 person-years) and comparing whether these are significantly different (95% CI and p-values) between men and women. This could be done for all-cause and cause-specific mortality. Any reference to "rates" in the paper as it stands is actually referring to hazards or risk, not rates.

6. The Kaplan Meier curves only show the unadjusted effects. Adjusted survival curves are important as well, especially given the different baseline characteristics between psychological distress yes/no and M/F.

7. Limitations should include the low response rate of the Scania public health survey (54.1%) and implications on the study findings; e.g. do we have information on non-respondents? %women/men, % with chronic or mental illnesses, etc. Also, what percentage of Scania's population is covered by the survey; who are excluded (those in hospitals, long term care, etc)?

*8. The discussion reiterates a lot of the results already presented in the text and tables. This section also introduces new analyses not mentioned in the methods or results. Once these are taken out or moved, the authors will have sufficient space to delve deeper into the interpretation and the "so-what" of their findings which is currently lacking in the discussion. For example, I would have liked to see more explanation around the gender paradox drawing from earlier literature of gender differences in psychological distress. What are the implications of these findings and what are the future directions?

Minor comments:

1. Abstract: mention GHQ-12 as the measure used to ascertain psychological distress.

2. Please provide rationale or references for why GHQ-12 cutpoints were used.

3. Report median follow-up time (in addition to or in place of maximum and mean).

	4. Please include more details about how the PH assumption was not violated in the Methods. 5. A few typos: continuous (p. 7), 5.3 years (p. 7)., " SES was defined by 12 categories *of employment*" (p. 7) 6. Were other SES measures (income, education) available? Why were these not examined and what was the rationale for focusing on employment? 7. Covariates: more information about the names of the registries/datasets and where other covariate information came from. Is it from the Scania public health survey? 8. p. 8: not typical for the Table numbers to be spelled out in the Methods; these can be removed. 9. p. 17. Not sure what "The loss of these participants in the present study is a selection bias in a weakening direction but of low magnitude on final results" means.
--	--

VERSION 1 – AUTHOR RESPONSE

Reviewer 1 (Andreas Lundin, Karolinska Institutet, PHS, Sweden):

The introduction

1. Reviewer: The GHQ-12 predicts all-cause mortality, and also death from cancer and CVD as expected from previous literature. This could be taken as evidence of predictive validity of the GHQ-12 in this setting, and because of the gender stratified analyses, that this is invariant across sex. What I think is missing is a sentence or two explaining why a measure of psychological distress would lead to these specific outcomes, rather than say depression or suicide. A mechanism could be presented for this (in the introduction).

Answer: We thank the reviewer for this comment. We have now rewritten this section, please see the first paragraph in the Introduction section.

2. Reviewer: The authors make the sex stratified analysis an important feature of their article, but this is not reflected in the title.

Answer: We thank the reviewer for this comment. The title has now been changed to "Poor psychological health and 5-year mortality: A population-based prospective cohort study stratified by gender in Scania, Sweden".

Methods

3. Reviewer: A design weight corrected for non-participation is presented, but a bit too brief. I suggest the construction of the weights is presented in a bit more detail. The strata are geographical areas (municipalities and city areas) so a weight to compensate for different sample probabilities for these are probably the first weight. This is the corrected for non-participation by using register information on all those in the sample frame. Relatedly, the use of this weight should be mentioned in the statistical analysis section.

Answer: We thank the reviewer for this comment. To compensate for selection bias, the geographically stratified random sample was weighted by age, sex, country of birth, marital status, income, and education through a weighting variable designed by Statistics Sweden. This has now been clarified under Participants and study design.

4. Reviewer: Last sentence in participants section: those 136 most likely emigrated or were dead? Please clarify.

Answer: We thank the reviewer for this comment. The 136 persons were most likely not dead, as all deaths of Swedish citizens, whether nationally or abroad, would have been recorded in the death register by The National Board of Welfare. For some reason, these 136 individuals were not traceable in the registers of health care during the follow-up period of more than 5 years. We can only speculate that these individuals could have lived abroad and therefore not been in touch with Swedish health care.

5. Reviewer: Under description of the GHQ-12 you describe that scores were used to construct zones. Could you describe why these specific cut offs were used? Here it would also be good to add Cronbach coefficient alpha for the GHQ-12, separately for men and women to show if the scale had the same properties for men and women. Any differences in predictive ability could be due to gender invariance.

Answer: We thank the reviewer for this comment. The GHQ-12 score was divided into four categories based on previous literature in order to investigate a dose-response association between GHQ-12 and mortality. The rationale for our division was that the majority (66 %) scored zero (69 % men; 63 % women), sub-clinical distress was by the Swedish conventional definition a score below 3, i.e. 1-2 (16 % scored 1-2, 15 % men; 17 % women), and the most even distribution of the remaining population was at cut-off 5/6 (9 % were defined as moderately and highly distressed respectively, 8 % + 8% of men and 10 % + 11 % of women). Reliability was good with Cronbach coefficient alpha 0.897 for men and 0.894 for women.

Discussion

6. Reviewer: comparison with previous literature is made with Finland. Finland is not a Scandinavian county as stated.

Answer: We thank the reviewer for this comment and this fault has now been corrected.

Tables

7. Reviewer: tables in the text are good. The tables following the article, numbered in a similar way seem to test a different cut off score (four or more symptoms). Is this correct?

Answer: We thank the reviewer for this comment. Yes, this is correct! The comparison between the two different cut-off scores has now been clarified as a minor aim. Supplementary Tables 1-4 (GHQ \geq 4) show identical analyses to Tables 1-4 (GHQ \geq 3). The reason for the comparison is that a recent Swedish case-control study concluded that the best sensitivity and specificity of GHQ-12 was seen at cut-off \geq 4 when discriminating between healthy controls and psychiatric outpatients. We were curious as to how results of the present study would be affected by a higher GHQ-12 cut-off.

Reviewer 2 (Alexander M/Ponizovsky, Ministry of Health, Israel):

This study using the representative sample of the general population in the Swedish region of Scania have replicated previous findings on the well-established relationships between the GHQ-12 psychological distress scores and all-cause and specific-cause mortality rates. In addition, gender differences were shown in the psychological distress-mortality relationships with higher distress scores in women, but higher mortality rates in men.

1. Reviewer: The study methodology is adequate, although the moderating/mediating effects of gender on the psychological distress-mortality association would be additionally explored.

Answer: We thank the reviewer for this comment. We have now added a test for effect modification by gender as stated in the second sentence under Statistics: "As the main aim of the study was to test gender differences, an initial test for effect modification by gender was performed. The interaction term between psychological distress and sex was significant (Hazard Ratio (HR) = 0.6; P = 0.002) indicating that the effect of psychological distress on mortality was different for men and women."

Reviewer: Results of the study are presented clearly. Limitations of the study are generally noted.

2. Reviewer: Because the main aim of the study was declared as to indicate gender differences in the distress-mortality relationships, the Introduction section needs more theoretical background on biological/psychological/social differences in psychological morbidity and distress expression in men and women.

Answer: We thank the reviewer for this comment. The Introduction section has been largely rewritten and now includes more theoretical background.

3. Reviewer: Also, in the Discussion section, it is absolutely insufficient to say the phrase “Among the possible explanations of the gender paradox are differentials between the sexes in biological risk, in health behaviors and social roles, and in health seeking behavior”. “The possible explanations” should be not only listened, but also discussed in details and they should be supported by the findings of previous studies and the above-mentioned theoretical background.

Answer: We thank the reviewer for this comment. We have thoroughly rewritten both the Introduction and the Discussion sections based on your good advice.

Reviewer 3 (Maria Chiu, University of Toronto):

Thank you for the opportunity to review this interesting and well-written paper entitled, "Psychological distress measured by GHQ-12 and mortality: A 5-year prospective population-based study in Scania, Sweden." The aim of this paper was to investigate whether the association between psychological distress and mortality differed between men and women in Scania, Sweden. The authors studied 28198 adults followed for up to 5 years and found that those who reported psychological distress at baseline had a higher hazards of dying than those with no psychological distress and this effect differed between men and women. They describe a morbidity-mortality gender paradox, in which women had higher prevalence of psychological distress but men with psychological distress have a higher risk of mortality.

Major comments (*most important to address):

Methods:

1a*. Reviewer: It is not clear whether weighted analyses were performed for this study. If the intention was to generalize these findings to the population of Scania and adjust at least in part for differential survey sampling, then weighted analyses would be recommended. Specifically, all statistical tests (i.e. t-tests, chi-square tests, interaction testing, Cox PH, etc.) would have to be weighted.

Answer: We thank the reviewer for this comment. All analyses have been performed on weighted data which is now stated in the Statistics section.

1b*. Reviewer: Furthermore, the authors should describe what methods were used to ensure that appropriate variance estimations were performed with weighted data (e.g. bootstrapped p-values, 95% confidence intervals?)

Answer: We thank the reviewer for this comment. Appropriate variance estimations were performed by bootstrap method (2000 replicates), 95% confidence intervals, which is now specified in the statistics section.

2. Reviewer: What did you mean by "The differences between unweighted and weighted data were small". Which variables were examined?

Answer: We thank the reviewer for this comment. Overall, the differences were small regarding prevalence for practically all included variables regardless of whether data was weighted or not. However, all analyses in the revised manuscript have now been performed on weighted data as recommended.

Statistics

3*. Reviewer: If the main aim of the study was to test gender differences, was there an initial statistical test for effect modification by gender? I don't doubt that there's an interaction, but would be good to describe in methods and state that the interaction term was significant in model 0, 1, 2 (p=)?

Answer: We thank the reviewer for this comment. We have now added the following under Statistics: "As the main aim of the study was to test gender differences, an initial test for effect modification by gender was performed. The interaction term between psychological distress and sex was significant (Hazard Ratio (HR) = 0.6; P = 0.002) indicating that the effect of psychological distress on mortality was different for men and women."

Results

4. Reviewer: it would be good to add more details describing findings from Table 1; i.e. factors that are present in those with psychological distress in men vs. women.

Answer: We thank the reviewer for this comment. We have now added this information.

5*. Reviewer: One major concern in the interpretation of the gender differences is whether they are statistically different. The 95% CIs for HRs, for example, overlap between men and women. I strongly suggest calculating and reporting the *mortality rates* (# of deaths per 100 person-years) and comparing whether these are significantly different (95% CI and p-values) between men and women. This could be done for all-cause and cause-specific mortality. Any reference to "rates" in the paper as it stands is actually referring to hazards or risk, not rates.

Answer: We thank the reviewer for this comment. We have now calculated mortality rates for all-cause and cause-specific mortality. As can be seen from the tables below, the male to female incidence rate ratios (IRR) show statistically significant differences between men and women for all mortality outcomes ($p < 0.001$ for all outcomes except cancer mortality $p = 0.013$). The 95% CIs for incidence rates per 100 person-years 95% overlapped minimally for men and women in cancer mortality and were non-overlapping for all other mortality outcomes.

Risk of overall mortality.					
	No. of Death	No. of Person-Years	Incidence Rate per 100 Person-Years.	Rate (95 % CI)	Male to female Incidence Rate Ratio (95% CI; P-value)
Male	464	59 625	0.78	0.71 - 0.85	1.83 (1.58-2.11; <.001)
Female	311	73 021	0.43	0.38 - 0.48	

Risk of overall Cardiovascular mortality.					
	No. of Death	No. of Person-Years	Incidence Rate per 100 Person-Years.	Rate (95 % CI)	Male to female Incidence Rate Ratio (95% CI; P-value)
Male	163	59 625	0.27	0.23 - 0.32	2.98 (2.24-3.96; <.001)
Female	67	73 021	0.09	0.07 - 0.12	

Risk of overall Cancer mortality.					
	No. of Death	No. of Person-Years	Incidence Rate per 100 Person-Years.	Rate (95 % CI)	Male to female Incidence Rate Ratio (95% CI; P-value)

Male	168	59 625	0.28	0.24	-	0.33	1.32 (1.06-1.64; 0.013)
Female	156	73 021	0.21	0.18	-	0.25	

Risk of mortality due to other causes.							
	No. of Death	No. of Person-Years	Incidence Rate per 100 Person-Years.	Rate	(95 % CI)	Rate Ratio (95% CI; P-value)	Male to female Incidence Rate Ratio (95% CI; P-value)
Male	133	59 625	0.22	0.18	0.26	1.85 (1.41-2.42; <.001)	
Female	88	73 021	0.12	0.10	0.15		

6. Reviewer: The Kaplan Meier curves only show the unadjusted effects. Adjusted survival curves are important as well, especially given the different baseline characteristics between psychological distress yes/no and M/F.

Answer: We thank the reviewer for this comment. We have now clarified that the survival curves are adjusted for age.

7. Reviewer: Limitations should include the low response rate of the Scania public health survey (54.1%) and implications on the study findings; e.g. do we have information on non-respondents? %women/men, % with chronic or mental illnesses, etc. Also, what percentage of Scania's population is covered by the survey; who are excluded (those in hospitals, long term care, etc)?

Answer: We thank the reviewer for this comment. A total of 52142 persons aged 18–80 years (a random stratified sample selected from the official population registers of people living in Scania including 5.8% of the total population 18-80 years old) were invited to participate in the Scania public health survey 2008. The response rate of 54.1% is in line with those normally obtained at this time in this type of population-based public health surveys in Sweden and Europe. However, we have now specifically mentioned the response rate of the survey as a limitation and stated that non-responders were more often young, male, low-educated or foreign-born. We do not know if non-responders had more psychological distress than responders. There were probably more non-responders among those with serious mental disorder (such as psychosis, schizophrenia, severe depression) than in the general population due to the mental task involved in answering a large survey with 134 main questions (totalling 273 items including subqueries and follow-up questions). As the surveys were posted to the residential address specified in the official population registers of people living in Scania, we do not know how many people hospitalized at the time had the opportunity to answer the survey (they could if the survey was brought to them by their spouse).

Discussion

8*. Reviewer : The discussion reiterates a lot of the results already presented in the text and tables. This section also introduces new analyses not mentioned in the methods or results. Once these are taken out or moved, the authors will have sufficient space to delve deeper into the interpretation and the “so-what” of their findings which is currently lacking in the discussion. For example, I would have liked to see more explanation around the gender paradox drawing from earlier literature of gender differences in psychological distress. What are the implications of these findings and what are the future directions?

Answer: We thank the reviewer for this comment. We have now completely rewritten the discussion based on your good advice.

Minor comments:

Abstract

1. Reviewer: mention GHQ-12 as the measure used to ascertain psychological distress.

Answer: We thank the reviewer for this comment. This has now been clarified in the abstract.

Methods

2. Reviewer: Please provide rationale or references for why GHQ-12 cutpoints were used.

Answer: We thank the reviewer for this comment. The two cut-off points GHQ-12 ≥ 3 (conventionally used in Sweden) and GHQ-12 ≥ 4 (widely used internationally) are based on the literature in public health. A minor aim of the present study was to compare these two cut-off points

(results for GHQ-12 ≥ 3 shown in Tables 1-3 and results for GHQ-12 ≥ 4 shown in Supplementary Tables 1-3). The GHQ-12 score was furthermore divided into four categories (also based on previous literature) in order to investigate a dose-response association (results shown in Table 4 and Supplementary Table 4). The rationale for our division was that the majority (66 %) scored zero, sub-clinical distress was by definition in the Swedish context a score below 3, i.e. 1-2 (16 %), and the most even distribution of the remaining population was at cut-off 5/6, i.e. moderate distress 3-5 (9 %) and high distress 6-12 (9%). The proportions regarding an international context were in our data: no distress (66%), sub-clinical distress 1-3 (20%), moderate distress 4-6 (7%) and high distress 7-12 (7%).

3. Reviewer: Report median follow-up time (in addition to or in place of maximum and mean).

Answer: We thank the reviewer for this comment. *Mortality* was assessed from August 27, 2008

to December 31, 2013, which amounts to a maximum period of 5.34 years. Mean value men: 5.16 years; women: 5.21 years. Median value men: 5.28 years; women: 5.28 years. This has now been clarified in the last paragraph under Predictor and outcome variables.

4. Reviewer: Please include more details about how the PH assumption was not violated in the Methods.

Answer: We thank the reviewer for this comment. The proportional hazards assumption was considered fulfilled after inspection of the survival curves according to psychological distress (yes/no). Additional statistical tests on our study data indicated absence of perfect proportionality with regard to poor GHQ-12 and mortality across the 5.3-year period (see Supplemental file on PH assumption). This is in line with other studies showing a dilution over time for mortality impact of baseline psychological distress – see discussion under Strengths and limitations on page 20-21.

5. Reviewer: A few typos: continuous (p. 7), 5.3 years (p. 7), " SES was defined by 12 categories *of employment*" (p. 7)

Answer: We thank the reviewer for this comment. These typos have now been corrected.

6. Reviewer: Were other SES measures (income, education) available? Why were these not examined and what was the rationale for focusing on employment?

Answer: We thank the reviewer for this comment. Employment was based on register data from Statistics Sweden (SCB) with internal missing 1.6%. Level of education was a self-reported variable

in the Scania Public Health Survey with internal missing 3.5%. We had no information on income. Thus our choice of employment as a SES measure.

7. Reviewer: Covariates: more information about the names of the registries/datasets and where other covariate information came from. Is it from the Scania public health survey?

Answer: We thank the reviewer for this comment. We have now clarified on page 9 under Covariates that the participants' age and SES were registry data from Statistics Sweden (SCB) while all other covariates were self-reported data from the Scania public health survey.

8. Reviewer: p. 8: not typical for the Table numbers to be spelled out in the Methods; these can be removed.

Answer: We thank the reviewer for this comment. However, for reasons of clarity we choose to keep the Table numbers in the Method section as we refer both to Tables and Supplementary Tables.

Discussion

9. Reviewer: p. 17. Not sure what “The loss of these participants in the present study is a selection bias in a weakening direction but of low magnitude on final results” means.

Answer: We thank the reviewer for this comment. We meant that the loss of participants with poor psychological health and higher risk of premature mortality would bias results towards the null if a selection bias of non-participation among persons with serious psychiatric disease had an impact on final results. As this sentence was not easy to understand, we have removed it from the revised manuscript.

Additional information regarding references

Due to the major revision of this manuscript **several new references have been added** while others have been removed. We have marked the new references in bold to make it easier for you to see which ones are new to the revised manuscript (Nr **3, 4, 17, 18, 20-26, 30, 31**).

References removed are listed below (numbered as in the previous manuscript):

- 4 Lindström M, Fridh M, Rosvall M. Economic stress in childhood and adulthood, and poor psychological health: Three life course hypotheses. *Psychiatry Res* 2014;215:386-93.
- 5 Lundin A, Hallgren M, Theobald H, *et al.* Validity of the 12-item version of the General Health Questionnaire in detecting depression in the general population. *Public Health* 2016;136:66-74. 14
 Russ TC, Kivimäki M, Morling JR, *et al.* Association between psychological distress and liver disease mortality: A meta-analysis of individual study participants. *Gastroenterology* 2015;148:958-66.
- 17 Bastos TF, Canesqui AN, Berti de Azavedo Barros M. "Healthy men" and high mortality: Contributions from a population-based study for the gender paradox discussion. *PLoS ONE* 2015;10(12):e0144520.
- 18 Rasul F, Stansfeld SA, Hart CL, *et al.* Psychological distress, physical illness and mortality risk. *J Psychosom Res* 2004;57:231-6.
- 19 Huppert FA, Whittington JE. Symptoms of psychological distress predict 7-year mortality. *Psychol Med* 1995;25(5):1073-86.
- 20 Batty GD, Hamer M, Der G. Does somatic illness explain the association between common mental disorder and elevated mortality? Findings from extended follow-up of study members in the UK Health and Lifestyle Survey. *J Epidemiol Community Health* 2012;66:647-9.
- 21 Pedersen SS, von Känel R, Tully PJ, *et al.* Psychosocial perspectives in cardiovascular disease. *Eur J Prev Cardiol* 2017;24:108-15.
- 22 Austad SN, Bartke A. Sex differences in longevity and in responses to anti-aging interventions: A mini-review. *Gerontology* 2016;62:40-6.
- 26 Virtanen M, Elovainio M, Josefsson, *et al.* Coronary heart disease and risk factors as predictors of trajectories of psychological distress from midlife to old age. *Heart* 2017;103:659-65.
- 27 Prince M, Patel V, Saxena S, *et al.* No health without mental health. *Lancet* 2007;370:859-77. 28
 Lumme S, Pirkola S, Manderbacka K, *et al.* Excess mortality in patients with severe mental disorders in 1996-2010 in Finland. *PLoS ONE* 2016;11(3):e0152223.

VERSION 2 – REVIEW

REVIEWER	Ponizovsky, Alexander M. Israeli Ministry of Health
REVIEW RETURNED	23-Sep-2021
GENERAL COMMENTS	I am satisfied with corrections made following my critical comments. I have no additional comments.
REVIEWER	Lyons, Ronan

	University of Wales Swansea, Swansea Clinical School
REVIEW RETURNED	06-Dec-2021

GENERAL COMMENTS	This is an interesting study which demonstrates that there is a relationship between scoring poorly on the GHQ12 and mortality in men but not women some 5.3 years later. It uses data from a population survey taken in 2008 with linked mortality records to the end of December 2013. The authors chose to analyse the data using a Cox proportional hazards model. Whilst the study appears to be well conducted and reported there are a couple of areas where I have concerns. In the methods section it is stated that "The proportional hazards assumption was considered fulfilled after inspection of the survival curves according to psychological distress". However, later there is a statement which contradicts this "Statistical tests on our study data indicated absence of perfect proportionality with regard to poor GHQ-12 and mortality across the 5.3-year period (see Supplemental file on PH assumption)" and those investigations do demonstrate deviation from the assumptions of the model. Given this is the case why did the authors choose such a model? There are other alternatives such as accelerated failure time models or simple logistic regression given that the follow up period was the same for all. Age was included as a continuous variable. Was the relationship linear? Table 2 shows attenuation of hazard ratios as more explanatory variables are included in the model. However, interaction term between male/female are not included but reference is made earlier to these being significant. This being the case means that the sex specific HRs are not interpretable as shown. Table 3 improves the analysis by calculating sex specific effect sizes and confidence intervals in fully adjusted models which show that there is no significant associations between psychological distress and mortality in women and the degree of association decreases in men but is still substantial. Table 4 explores a dose response relationship between four categories of psychological distress and mortality and demonstrated this clearly in men and not in women. However, given the very high survival rates in women power to detect any relationship between psychological distress and mortality will be very low. The linked mortality data are from 2013 which is 7-8 years ago. Are more up to date data available to improve the power of the analysis in women? The abstract states that "GHQ12 could potentially be used as one of several predictors of mortality, especially for men. In the future, screening tools for psychological distress should be validated for both men and women". Why especially for men when no association is shown for women? Perhaps delete especially.
---

VERSION 2 – AUTHOR RESPONSE

Reviewer: 1

Dr. Alexander M. Ponizovsky, Israeli Ministry of Health

Comments to the Author:

I am satisfied with corrections made following my critical comments.

I have no additional comments.

Answer: We thank the reviewer for this comment.

Reviewer: 2

Prof. Ronan Lyons, University of Wales Swansea

Comments to the Author:

This is an interesting study which demonstrates that there is a relationship between scoring poorly on the GHQ12 and mortality in men but not women some 5.3 years later. It uses data from a population survey taken in 2008 with linked mortality records to the end of December 2013. The authors chose to analyse the data using a Cox proportional hazards model. Whilst the study appears to be well conducted and reported there are a couple of areas where I have concerns.

1_Reviewer: In the methods section it is stated that “The proportional hazards assumption was considered fulfilled after inspection of the survival curves according to psychological distress”. However, later there is a statement which contradicts this “Statistical tests on our study data indicated absence of perfect proportionality with regard to poor GHQ-12 and mortality across the 5.3-year period (see Supplemental file on PH assumption)” and those investigations do demonstrate deviation from the assumptions of the model. Given this is the case why did the authors choose such a model? There are other alternatives such as accelerated failure time models or simple logistic regression given that the follow up period was the same for all.

Answer: We thank the reviewer for this comment. We have now chosen to reperform all statistical analyses by logistic regression analysis instead of Cox regression survival analysis, furthermore with 8.3 years follow-up instead of 5.3 years follow-up.

In our previous manuscript we considered the proportional hazard assumption to be adequately fulfilled by ocular inspection of the survival curves according to psychological distress (yes/no). As we argued in the Discussion section a certain dilution of effect by psychological distress on mortality can be expected over time (this has been shown e.g by Lee and Singh in their paper from 2020: Psychological distress and heart disease mortality in the United States: Results from the 1997-2014 NHIS-NDI Record Linkage Study.) In our literature search we also found that Cox regression analysis was clearly the method of choice as this was used by all previous comparable population-based studies on GHQ-12 and mortality.

After your feed-back we compared results from Cox regression analysis and logistic regression analysis (in tables 2-3) for both time-frames (5.3 years and 8.3 years, respectively). The effect measures and 95 % confidence intervals were fairly similar for these two methods, see below.

Table 2 with 5.3 years follow-up: a) HR by Cox regression analysis and b) ORs by Logistic regression analysis

Table 2a. HRs from Cox regression models for all-cause mortality and cause-specific mortality, showing association with psychological distress (GHQ \geq 3).

The 2008 Scania public health survey **with 5.3 years follow-up.**

Men and women combined, n=25503.

Cause of death	Model 0		Model 1		Model 2		Number of deaths
	HR	(95%CI)	HR	(95%CI)	HR	(95%CI)	
All causes	2.9***	(2.4-3.6)	2.1***	(1.7-2.7)	1.8***	(1.5-2.3)	775
Cause-specific:							
Cardiovascular	3.2***	(2.2-4.7)	2.4***	(1.6-3.6)	2.0***	(1.3-3.0)	230
Cancer	2.6***	(1.9-3.6)	2.0***	(1.4-2.8)	1.7**	(1.2-2.5)	324
Other causes	3.1***	(2.1-4.6)	2.1***	(1.4-3.1)	1.7*	(1.1-2.7)	221

Model 0 adjusted for age and gender.

Model 1 furthermore adjusted for socioeconomic status, physical activity, smoking, and alcohol.

Model 2 furthermore adjusted for chronic disease.

Significance levels: * p<0.05, ** p<0.01, *** p<0.001

Weighted Hazard Ratios. Bootstrap method (2000 replicates) for variation estimation.

Table 2b. ORs from Logistic regression models for all-cause mortality and cause-specific mortality, showing association with psychological distress (GHQ \geq 3).

The 2008 Scania public health survey **with 5.3 years follow-up.**

Men and women combined, n=25503.

Cause of death	Model 0		Model 1		Model 2		Number of deaths
	OR	(95%CI)	OR	(95%CI)	OR	(95%CI)	
All causes	3.1***	(2.5-4.0)	2.3***	(1.8-2.9)	1.9***	(1.5-2.5)	775
Cause-specific:							
Cardiovascular	3.1***	(2.1-4.7)	2.3***	(1.5-3.6)	1.9***	(1.3-3.0)	230
Cancer	2.5***	(1.8-3.5)	1.9***	(1.4-2.8)	1.7**	(1.2-2.4)	324
Other causes	2.9***	(1.9-4.5)	2.0**	(1.3-3.0)	1.7*	(1.1-2.6)	221

Model 0 adjusted for age and gender.

Model 1 furthermore adjusted for socioeconomic status, physical activity, smoking, and alcohol.

Model 2 furthermore adjusted for chronic disease.

Significance levels: * p<0.05, ** p<0.01, *** p<0.001

Weighted Odds Ratios. Bootstrap method (2000 replicates) for variation estimation.

Table 3 with 5.3 years follow-up: a) HR by Cox regression analysis and b) ORs by Logistic regression analysis

Table 3a HRs from Cox regression models for all-cause mortality and cause-specific mortality, showing association with psychological distress (GHQ \geq 3).

The 2008 Scania public health survey **with 5.3 years follow-up.**

Stratified by gender, n = 13984 women and 11519 men.

Cause of death	Model 0		Model 1	(95%CI)	Model 2		Number of deaths
	HR	(95%CI)	HR		HR	(95%CI)	
All causes							
Women	2.1***	(1.5-2.8)	1.4	(1.0-1.9)	1.2	(0.9-1.6)	311
Men	3.6***	(2.7-4.7)	2.8***	(2.1-3.8)	2.4***	(1.8-3.2)	464
Cause-specific:							
Cardiovascular							
Women	2.1*	(1.0-4.4)	1.2	(0.6-2.5)	1.1	(0.5-2.2)	67
Men	3.7***	(2.4-5.9)	3.1***	(1.8-5.1)	2.5***	(1.6-4.2)	163
Cancer							
Women	1.6*	(1.0-2.6)	1.2	(0.7-1.9)	1.0	(0.6-1.7)	156
Men	3.6***	(2.4-5.5)	3.1***	(1.9-4.8)	2.6***	(1.7-4.1)	168
Other causes							
Women	2.7*	(1.4-5.0)	1.7	(1.0-3.1)	1.4	(0.8-2.6)	88
Men	3.3***	(2.0-5.6)	2.3**	(1.3-4.0)	2.0*	(1.1-3.6)	133

Model 0 adjusted for age and gender.

Model 1 furthermore adjusted for socioeconomic status, physical activity, smoking, and alcohol.

Model 2 furthermore adjusted for chronic disease. Significance levels: * p<0.05, ** p<0.01, *** p<0.001

Weighted Hazard Ratios. Bootstrap method (2000 replicates) for variation estimation.

Table 3b ORs from logistic regression models for all-cause mortality and cause-specific mortality, showing association with psychological distress (GHQ \geq 3).

The 2008 Scania public health survey **with 5.3 years follow-up**.

Stratified by gender, n = 13984 women and 11519 men.

Cause of death	Model 0		Model 1		Model 2		Number of deaths
	OR	(95%CI)	OR	(95%CI)	OR	(95%CI)	
All causes							
Women	2.1** *	(1.5-3.0)	1.4	(1.0-1.9)	1.2	(0.8-1.7)	311
Men	4.1** *	(3.0-5.6)	3.2***	(2.3-4.4)	2.6***	(1.9-3.7)	464
Cause-specific:							
Cardiovascular							
Women	2.1	(1.0-4.3)	1.2	(0.6-2.4)	1.0	(0.5-2.1)	67
Men	3.7** *	(2.2-6.1)	3.1***	(1.8-5.3)	2.5***	(1.5-4.2)	163
Cancer							
Women	1.6	(1.0-2.5)	1.1	(0.7-1.9)	1.0	(0.6-1.6)	156
Men	3.5** *	(2.2-5.4)	2.9***	(1.8-4.7)	2.5***	(1.6-4.0)	168
Other causes							
Women	2.6*	(1.4-5.0)	1.7	(0.9-3.1)	1.4	(0.8-2.6)	88
Men	3.1** *	(1.8-5.4)	2.2**	(1.2-3.9)	1.9	(1.0-3.5)	133

Model 0 adjusted for age and gender.

Model 1 furthermore adjusted for socioeconomic status, physical activity, smoking, and alcohol.

Model 2 furthermore adjusted for chronic disease.

Significance levels: * p<0.05, ** p<0.01, *** p<0.001

Weighted Odds Ratios. Bootstrap method (2000 replicates) for variation estimation.

Table 2 with 8.3 years follow-up: a) HR by Cox regression analysis and b) ORs by Logistic regression analysis

Table 2a. HRs from Cox regression models for all-cause mortality and cause-specific mortality, showing association with psychological distress (GHQ \geq 3).

The 2008 Scania public health survey **with 8.3 years follow-up.**

Men and women combined, n=25503.

Cause of death	Model 0		Model 1		Model 2		Number of deaths
	HR	(95%CI)	HR	(95%CI)	HR	(95%CI)	
All causes	2.5***	(2.1-3.0)	1.9***	(1.6-2.3)	1.6***	(1.4-2.0)	1389
Cause-specific:							
Cardiovascular	3.1***	(2.3-4.1)	2.2***	(1.6-3.1)	1.9***	(1.4-2.6)	425
Cancer	1.9***	(1.4-2.5)	1.6***	(1.2-2.1)	1.4*	(1.0-1.9)	539
Other causes	2.9***	(2.2-3.9)	2.0***	(1.5-2.7)	1.7***	(1.3-2.3)	425

Model 0 adjusted for age and gender.

Model 1 furthermore adjusted for socioeconomic status, physical activity, smoking, and alcohol.

Model 2 furthermore adjusted for chronic disease.

Significance levels: * p<0.05, ** p<0.01, *** p<0.001

Weighted Hazard Ratios. Bootstrap method (2000 replicates) for variation estimation.

Table 2b. ORs from Logistic regression models for all-cause mortality and cause-specific mortality, showing association with psychological distress (GHQ \geq 3).

The 2008 Scania public health survey **with 8.3 years follow-up.**

Men and women combined, n=25503.

Cause of death	Model 0		Model 1		Model 2		Number of deaths
	OR	(95%CI)	OR	(95%CI)	OR	(95%CI)	
All causes	2.8***	(2.3-3.4)	2.1***	(1.7-2.6)	1.8***	(1.4-2.2)	1389
Cause-specific:							
Cardiovascular	2.9***	(2.1-4.1)	2.2***	(1.5-3.1)	1.8***	(1.3-2.6)	425
Cancer	1.7***	(1.3-2.3)	1.5*	(1.1-2.0)	1.3	(1.0-1.8)	539
Other causes	2.8***	(2.0-3.8)	1.9***	(1.4-2.6)	1.6**	(1.2-2.2)	425

Model 0 adjusted for age and gender.

Model 1 furthermore adjusted for socioeconomic status, physical activity, smoking, and alcohol.

Model 2 furthermore adjusted for chronic disease.

Significance levels: * p<0.05, ** p<0.01, *** p<0.001

Weighted Odds Ratios. Bootstrap method (2000 replicates) for variation estimation.

Table 3 with 8.3 years follow-up: a) HR by Cox regression analysis and b) ORs by Logistic regression analysis

Table 3a HRs from Cox regression models for all-cause mortality and cause-specific mortality, showing association with psychological distress (GHQ \geq 3).

The 2008 Scania public health survey **with 8.3 years follow-up**.

Stratified by gender, n = 13984 women and 11519 men.

Cause of death	Model 0		Model 1		Model 2		Number of deaths
	HR	(95%CI)	HR	(95%CI)	HR	(95%CI)	
All causes							
Women	2.1***	(1.7-2.7)	1.5***	(1.2-2.0)	1.3*	(1.1-1.7)	574
Men	2.9***	(2.3-3.6)	2.2***	(1.7-2.9)	1.9***	(1.5-2.4)	815
Cause-specific:							
Cardiovascular							
Women	2.6***	(1.6-4.3)	1.7*	(1.1-2.8)	1.5	(0.9-2.4)	140
Men	3.2***	(2.2-4.7)	2.5***	(1.6-3.8)	2.1***	(1.4-3.2)	285
Cancer							
Women	1.6*	(1.1-2.4)	1.3	(0.9-1.9)	1.2	(0.8-1.8)	258
Men	2.2***	(1.5-3.2)	1.9**	(1.2-2.9)	1.6*	(1.1-2.5)	281
Other causes							
Women	2.5***	(1.6-3.9)	1.7*	(1.1-2.6)	1.4	(0.9-2.1)	176
Men	3.2***	(2.2-4.7)	2.3***	(1.5-3.4)	1.9**	(1.3-2.9)	249

Model 0 adjusted for age and gender.

Model 1 furthermore adjusted for socioeconomic status, physical activity, smoking, and alcohol.

Model 2 furthermore adjusted for chronic disease. Significance levels: * p<0.05, ** p<0.01, *** p<0.001

Weighted Hazard Ratios. Bootstrap method (2000 replicates) for variation estimation.

Table 3b ORs from logistic regression models for all-cause mortality and cause-specific mortality, showing association with psychological distress (GHQ≥3).

The 2008 Scania public health survey **with 8.3 years follow-up**.

Stratified by gender, n = 13984 women and 11519 men.

Cause of death	Model 0		Model 1		Model 2		Number of deaths
	OR	(95%CI)	OR	(95%CI)	OR	(95%CI)	
All causes							
Women	2.3** *	(1.7-3.0)	1.6**	(1.2-2.1)	1.4*	(1.0-1.8)	574
Men	3.3** *	(2.5-4.4)	2.6***	(1.9-3.5)	2.1***	(1.6-2.9)	815
Cause-specific:							
Cardiovascular							
Women	2.5** *	(1.5-4.2)	1.7	(1.0-2.9)	1.4	(0.8-2.4)	140
Men	3.1** *	(2.1-4.8)	2.5***	(1.5-4.0)	2.1**	(1.3-3.3)	285
Cancer							
Women	1.5*	(1.0-2.3)	1.2	(0.8-1.9)	1.1	(0.7-1.7)	258
Men	2.0** *	(1.3-3.0)	1.7*	(1.1-2.7)	1.5	(1.0-2.3)	281
Other causes							
Women	2.4** *	(1.6-3.8)	1.6*	(1.0-2.5)	1.3	(0.8-2.1)	176
Men	3.0** *	(2.0-4.5)	2.2***	(1.4-3.3)	1.8*	(1.1-2.8)	249

Model 0 adjusted for age and gender.

Model 1 furthermore adjusted for socioeconomic status, physical activity, smoking, and alcohol.

Model 2 furthermore adjusted for chronic disease.

Significance levels: * $p < 0.05$, ** $p < 0.01$, *** $p < 0.001$

Weighted Odds Ratios. Bootstrap method (2000 replicates) for variation estimation.

2. Reviewer: Age was included as a continuous variable. Was the relationship linear?

Answer: We thank the reviewer for this comment. The relationship between age and mortality was not linear but exponential. The mortality risk was approximately 100 times higher in the oldest group compared with the youngest in our data, which mirrors the relationship between age and mortality in the general population.

3. Reviewer: Table 2 shows attenuation of hazard ratios as more explanatory variables are included in the model. However, interaction term between male/female are not included but reference is made earlier to these being significant. This being the case means that the sex specific HRs are not interpretable as shown. Table 3 improves the analysis by calculating sex specific effect sizes and confidence intervals in fully adjusted models which show that there is no significant association between psychological distress and mortality in women and the degree of association decreases in men but is still substantial.

Answer: We thank the reviewer for this comment. The intended logic of Table 2 and Table 3 was to first present data on men and women combined with adjustments for age and sex (as in most comparable studies) and then present separate effect measures for men and women by gender stratification (an extension of previous research).

4. Reviewer: Table 4 explores a dose response relationship between four categories of psychological distress and mortality and demonstrated this clearly in men and not in women. However, given the very high survival rates in women power to detect any relationship between psychological distress and mortality will be very low. The linked mortality data are from 2013 which is 7-8 years ago. Are more up to date data available to improve the power of the analysis in women?

Answer: We thank the reviewer for this comment. When we first submitted this manuscript in 2018 we had access to mortality data with follow-up 5.3 years. The mortality data has now been extended to a follow-up 8.3 years.

5. Reviewer: The abstract states that “GHQ12 could potentially be used as one of several predictors of mortality, especially for men. In the future, screening tools for psychological distress should be validated for both men and women”. Why especially for men when no association is shown for women? Perhaps delete especially.

Answer: We thank the reviewer for this comment. The word “especially” is now motivated by the weaker association found in women in comparison to men.